# Solar and wind energy enhances drought resilience and groundwater sustainability

Xiaogang He [1,2,3]*, Kairui Feng [1], Xiaoyuan Li[1], Amy B. Craft[4], Yoshihide Wada [3,5], Peter Burek [3], Eric F. Wood [1] & Justin Sheffield[1,6]*

Water scarcity brings tremendous challenges to achieving sustainable development of water resources, food, and energy security, as these sectors are often in competition, especially during drought. Overcoming these challenges requires balancing trade-offs between sectors and improving resilience to drought impacts. An under-appreciated factor in managing the water-food-energy (WFE) nexus is the increased value of solar and wind energy (SWE). Here we develop a trade-off frontier framework to quantify the water sustainability value of SWE through a case study in California. We identify development pathways that optimize the economic value of water in competition for energy and food production while ensuring sustainable use of groundwater. Our results indicate that in the long term, SWE penetration creates beneficial feedback for the WFE nexus: SWE enhances drought resilience and benefits groundwater sustainability, and in turn, maintaining groundwater at a sustainable level increases the added value of SWE to energy and food production.

[1] Department of Civil and Environmental Engineering, Princeton University, Princeton, NJ, USA. [2] Water in the West, Woods Institute for the Environment, Stanford University, Stanford, CA, USA. [3] International Institute for Applied Systems Analysis, Laxenburg, Austria. [4] Woodrow Wilson School of Public and International Affairs, Princeton University, Princeton, NJ, USA. [5] Department of Physical Geography, Utrecht University, Utrecht, Netherlands. [6] Geography and Environment, University of Southampton, Southampton, UK. *email: hexg@princeton.edu; justin.sheffield@soton.ac.uk

As the central element of the water–food–energy (WFE) nexus[1–3], effective management of water resources, especially for regulated river basins, is key to meet societal needs, including irrigation supply for food production and reservoir water release for hydropower generation. However, current water management strategies are often carried out independently for each sector, leading to competition for water resources[4,5]. This is likely to be exacerbated by the potential for increasing severity of drought under climate change[6] and growing demand for limited water resources[7,8]. For example, globally, 54% of hydropower plants compete with irrigation water use[9], and this competition between food production and hydropower generation has been increasing with several hot spots identified around the world. The competition usually happens between upstream and downstream sources. For instance, upstream hydroelectric power plants tend to store more water to increase and maintain the hydraulic head for power generation, even during the dry season. In contrast, downstream users need water released from upstream reservoirs to irrigate crops with a different timing (e.g., during the growing season). In some cases, a lack of available surface water puts a burden on groundwater, which also acts as a buffer to alleviate drought, leading to groundwater depletion[10–13], given the slow process of groundwater recharge to aquifers[14]. Meanwhile, increased water scarcity[15,16] and shifts in the timing of streamflow[17,18] could further strain the WFE nexus and exacerbate the conflicts or trade-offs between irrigation and hydropower. For instance, traditional reservoir operation rules without consideration of the non-stationarity[19] of hydroclimate may no longer be efficient enough to navigate the trade-offs due to the seasonal imbalance between water supply and demand.

Here, we argue that the water allocation trade-offs between hydropower generation and irrigation use, and their future evolution, can be potentially solved by consideration of integrated management tools and the fast increase of low-carbon energy generation, such as solar and wind energy (SWE). Given the fact that SWE deployment is accelerating and is particularly substitutable for hydropower if they are paired with energy storage facilities (e.g., thermal storage, batteries), energy systems are becoming less reliant on hydropower, as well as fossil fuels, especially for developed regions. Consequently, water used to drive turbines for hydropower generation can be saved for irrigation purposes to ensure food production, whilst reducing groundwater usage thereby increasing groundwater sustainability especially under drought. Here we emphasize the social value of SWE for environmental sustainability, which remains poorly understood in the scientific community and policy circles, through a case study in California. We first examine how water scarcity, as well as SWE, influences decisions surrounding the optimal and sustainable allocation of water for hydropower generation and food production. We then estimate the unrecognized and under-appreciated value of SWE beyond its role in the traditional energy sector and the synergies between SWE and groundwater to enhance drought resilience and environmental sustainability. Our analysis can help develop and integrate impact pathways into policy support for positive practical changes for sustainable water and food security.

## Results

### SWE enhances groundwater sustainability.
California recently endured a record-breaking drought after 2012 (refs. [20,21]), which significantly impacted food production[22], reduced hydropower generation[23] and caused severe environmental issues (e.g., groundwater depletion, wildfires, tree mortality, land subsidence). As the largest agricultural producing state in the USA, California earned ~$47 billion from its agricultural sector and contributed to

13% of the US total in 2015 even during the drought. The maintenance of crop revenue and overall resilience of the agricultural sector largely relied on the unsustainable groundwater overdraft, which effectively offset the drought impact, but contributed to severe groundwater depletion (~3.7 km³/year[24]). In the energy sector, during this driest year of the drought, decreased surface water availability sent the in-state hydropower generation plunging to 7% of the total electricity generated, substantially below the state's long-term average of around 18%[23]. This power deficit was offset by electricity generated through the rapidly growing solar and wind fleet, as well as from increased use of natural gas and electricity purchased from out-of-state sources[23]. Furthermore, for the first time, in 2012, solar and wind electricity generation exceeded hydropower in California[23] due to the declining cost of wind turbines and solar photovoltaic (PV) in conjunction with the popularity and stringency of the Renewables Portfolio Standard (RPS), which mandates a certain proportion of renewables in the energy production.

The penetration of SWE not only offset some of the decreases in hydropower but has implications beyond the energy sector given the inextricable links among food, energy and water. This added value can be derived by considering the sustainability trade-offs within the WFE nexus. In general, there is a direct trade-off between hydroelectricity production and irrigation of crops in how surface water is allocated between the two. There is also an indirect trade-off between hydroelectricity production and groundwater abstraction, as groundwater can substitute for reduced surface water availability during a drought, which in the case of the recent California drought allowed crop production to generally be unaffected. Given relatively low groundwater recharge rates and increasing risk of drought, this indirect trade-off highlights potential sustainability challenges for groundwater.

We adopt the trade-off frontier (TF) (also called production possibility frontier, see Methods for details) to investigate the compromise between hydroelectric generation and groundwater abstraction in California given a set of surface water constraints (gray solid lines in Fig. 1) varying from a dry to a wet year. We use a calibrated and physically based hydrological model with water management options to dynamically simulate the surface water availability for hydropower production as well as the irrigation water requirement (including both surface water and groundwater) for food production. We then estimate how surface water and groundwater can be optimally allocated to maximize the total economic revenue ($R$). Given the water constraint in a certain year, surface water allocation strategies are efficient if they fall along the TF curve, while they are inefficient/unattainable if strategies fall below/above the TF. A strategy is inefficient if surface water is not fully used for hydropower (production is lower than potential) and agriculture (irrigation is less than crop demand), and groundwater is used for irrigation instead. A strategy is unattainable if the water demand for both hydropower production and irrigation exceeds the surface water availability, and the shortfall in irrigation demand cannot be satisfied by the current groundwater abstraction rate. Iso-revenue curves (green dashed lines in Fig. 1) connect points of equal economic profit with different quantities of hydroelectricity production, economic cost of groundwater pumping and revenue loss due to crop failure (see Methods for details on revenue calculation). Crop revenue may be reduced if water demand is not met by surface water allocation and the current rate of groundwater abstraction. Iso-revenue curves are convex given the law of diminishing marginal utility. The point of tangency between the TF and the iso-revenue curve (black point in Fig. 1) indicates the optimal (or economically efficient) condition where efficient water allocation and maximum revenue could both be achieved through appropriate

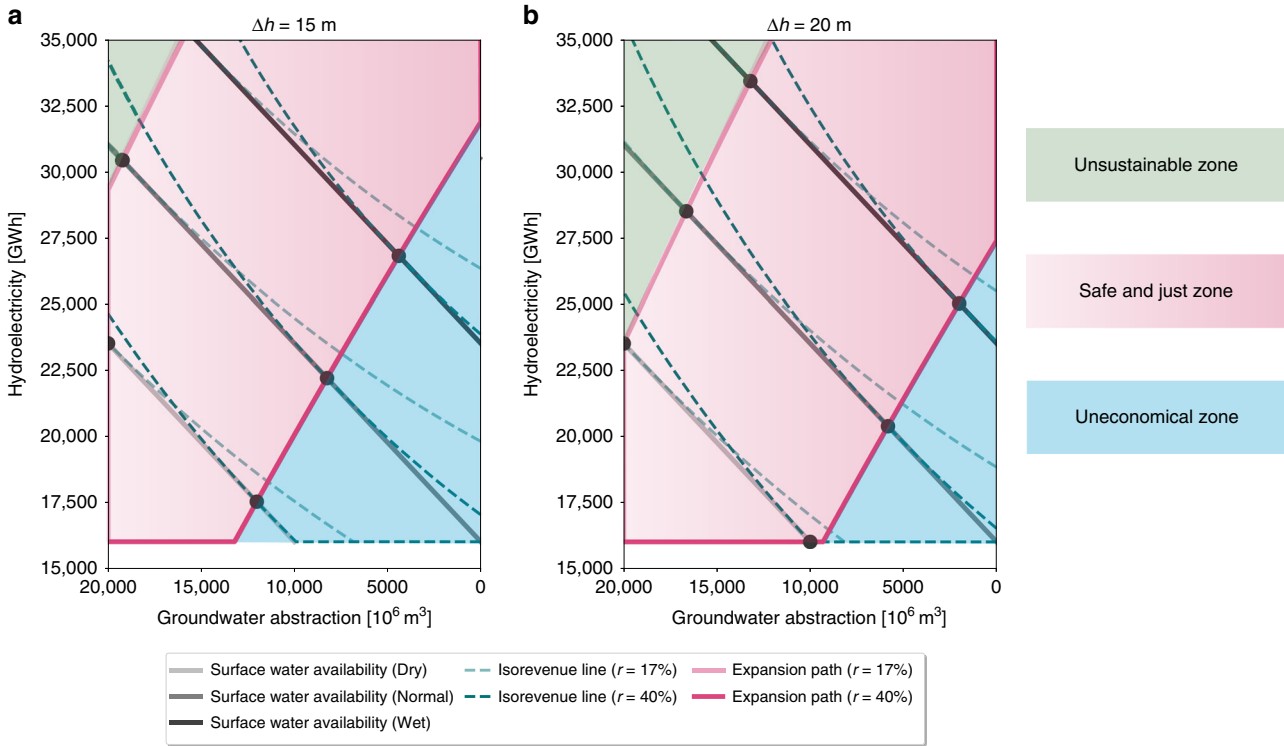

**Fig. 1** Trade-off analysis between groundwater abstraction and hydropower generation. Frontiers curves (gray solid lines) for three inflow availability conditions (dry, normal and wet year) during the historical period estimated from CWatM, the corresponding iso-revenue lines (green dashed lines) and expansion paths (pink solid lines) for optimal water allocation given current ($r = 17\%$) and future penetration of SWE ($r = 40\%$) under different groundwater pumping lift ($\Delta h$): **a** $\Delta h = 15$ m, **b** $\Delta h = 20$ m. Black dots represent the optimal condition, where surface water allocation is efficient and revenue can be maximized. Light-green shaded area represents unsustainable zone, where more groundwater is abstracted for irrigation as less surface water is available due to its use for hydroelectric generation. Pink shaded area is the safe and just zone, where optimal points can be achieved. Light-blue shaded area represents uneconomical zone, where we sacrifice the revenue from hydropower production in order to maintain the groundwater sustainability

policy instruments, such as SWE penetration and groundwater abstraction caps (as discussed later). Externalities or market failure may distort the iso-revenue curve, and social and technological constraints (e.g., cropping decisions, lack of infrastructure for water storage and diversion) may cause the allocation to be unattainable. On top of these factors, hydro-climate variability will shift the TF inward and outward for low (lower surface water availability in a dry year) and high inflow (higher surface water availability in a wet year) conditions, respectively, compared to the normal year. Connecting the optimal points under different surface water availability conditions forms a so-called expansion path (EP, pink lines in Fig. 1, see supplementary materials for algorithms applied to find EP). The EP informs policymaking by identifying the optimal water allocation to secure food production while balancing hydroelectric generation and groundwater abstraction as surface water availability changes.

To examine the added value of SWE in reducing sustainability trade-offs, we use this framework to quantify how optimal strategies maximizing hydroelectricity and agricultural income, whilst avoiding groundwater depletion, are altered by the penetration of SWE. California has seen sustained growth of solar and wind power, which account for 17% of statewide electricity generation in 2016 (data from California Energy Commission). By 2030, solar and wind are projected to generate 35–40% of total electricity[25] to achieve the goal of 50% renewables together with hydropower (State Bill No. 350). Given this target, we consider two penetration scenarios to examine how future penetration of SWE (40%) would influence the

hydroelectricity–groundwater trade-offs compared to the current situation (17%) under different surface water availability conditions. Penetration of SWE influences the shape and position of the iso-revenue lines and therefore changes the position of the optimal point (Fig. 1, see Methods for details). Iso-revenue lines in the current penetration scenario have smaller curvature than those in the future penetration scenario, indicating smaller marginal revenue of hydroelectricity. This implies that as more SWE is deployed and the hydroelectricity price goes down, to maintain the same revenue, one unit of abstraction of groundwater requires more hydroelectric generation to compensate the pumping cost. This in turn shifts the EP rightward (more sustainable for groundwater), favoring surface water allocation for irrigation and reducing groundwater abstraction. This happens because hydropower is displaced by solar and wind, surface water, which would otherwise generate hydroelectricity, is conserved and can now be used for irrigation. As indicated by the horizontal part of the EP, the initial allocation of surface water is targeted for crop production with higher priority until surface water availability surpasses a certain threshold. This is especially the case when surface water becomes scarcer during a drought, and the cost of pumping groundwater to the surface becomes higher than the revenue gained from hydroelectricity generation. As surface water becomes abundant, it starts to be allocated to both hydropower generation and irrigation with equal marginal water allocation efficiency as shown in the diagonal part of the EP.

The sustainability value of SWE is, however, tempered by groundwater depletion. If groundwater is abstracted at

unsustainable rates (abstraction exceeds recharge, as is currently happening in the Central Valley of California) then the value of SWE in reducing surface water allocation trade-offs also decreases. Groundwater depletion results in higher pumping lift ($\Delta h$) and costs, which therefore further exacerbates the trade-offs between groundwater abstraction and hydroelectric generation. Consequently, this pushes the socially optimal EP together with the safe and just zone further to the right (Fig. 1b compared to Fig. 1a) suggesting less groundwater is abstracted. However, given the increased groundwater depletion ($\Delta h = 20$ m), any additional groundwater abstraction could make the groundwater aquifer less sustainable (increased unstainable zone in Fig. 1b). We also note the enhanced length of the horizontal EP (Fig. 1b), which implies that groundwater depletion further reduces the marginal revenue of hydropower during drought periods. As groundwater becomes scarcer and more expensive, hydropower should be reduced to save water for irrigation. This leads to the shrinkage of the uneconomical zone, as groundwater pumping costs are saved with a higher magnitude compared to the magnitude of the revenue loss due to the reduced hydropower.

**Taking into account groundwater sustainability policies.** The TF–EP framework envisions the optimal pathway to balance the trade-offs between hydroelectric generation and groundwater abstraction, which in reality is over-optimistic and may not be achievable owing to a set of physical, political and economic constraints. One possible constraint comes from regulation policies, which could act as a barrier to achieving the social optimum, such as the recently passed Sustainable Groundwater Management Act (SGMA) in California. Such quantity-oriented regulations set the limit for groundwater abstraction ($g_w$) (see the schematic illustration in Fig. 2a), under which the optimal point can only fall into the hatched area. For years with relatively low surface water availability, this groundwater cap ($g_w^{\mathrm{Cap}}$) reduces efficiency even with high penetration of SWE, as the optimal condition is not attainable (that is, the optimal point $O_C/O_F$ moves to $A$). As water availability further increases, imposing limitations via regulations may not influence the optimal point under future penetration of SWE (as $O_F'$ is still in the hatched area), whereas under current penetration the optimal point is shifted from $O_C'$ to $A'$. Limiting groundwater use in turn increases the risk of crop failure and therefore reduces the crop revenue. To quantify this, we define the relative revenue loss ($\delta$) as: $\delta = 1 - \frac{R(g_w \leq g_w^{\mathrm{Cap}})}{R(g_w \leq \infty)}$, which has a range from 0 to 1, with 0 indicating zero revenue loss and the optimal point still achievable. Intuitively according to Fig. 2a, this implies that control on groundwater abstraction is not stringent enough to move the optimal point out of the hatched area, which means regulation policies do not exert any impacts on the optimal water allocation and therefore the total revenue is not influenced. As $\delta$ increases, we face higher revenue loss either because we have a stricter groundwater cap, or there is lower surface water availability. With fixed surface water availability (Fig. 2b), $\delta$ monotonically decreases as $g_w^{\mathrm{Cap}}$ increases for both current penetration ($P_C^1 \to P_C^2$) and future penetration ($P_F^1 \to P_F^2$). In other words, as we loosen the limit on groundwater use, the relative loss will be reduced. This further implies that groundwater sustainability is put at risk for economic revenue.

Groundwater pumping lift ($\Delta h$) adds another layer of complexity to the relative revenue loss ($\delta$). Higher pumping lift indicates higher pumping cost associated with more severe groundwater depletion. In the plane with zero revenue loss, higher pumping lift would require more stringent groundwater regulations ($P_C^2 \to P_C^3$) in order to achieve the optimal trade-offs

as described in Fig. 1. We note that future penetration of SWE pushes the boundary toward a smaller groundwater cap ($P_C^2 P_C^3$ is shifted to $P_F^2 P_F^3$), which implies that we can set relatively strict regulations for groundwater sustainability with a higher percentage mix of SWE in the energy portfolio. Ideally, society would like to move toward the black point in Fig. 2b with lower revenue loss and higher groundwater storage recovery (smaller pumping lift). Our results highlight the difficulty in recovering groundwater storage ($P_C^4 \to P_C^1$) once it has depleted to a certain extent even with extremely strict regulations (e.g., near zero allowance). This is because a small reduction of pumping lift (slightly recovery of groundwater storage) would result in a significant increase in revenue loss (i.e., $\delta$ increases dramatically along the direction of $P_C^4 \to P_C^1$ with a slight decrease of $\Delta h$). As $\delta$ is defined as a relative term, it shows how much total revenue will be reduced if groundwater regulation policy is added as a constraint relative to the situation without such constraint. Given that people tend to be loss averse (e.g., minimize relative revenue losses compared to the optimal situation) and prefer to make decisions based on losses rather than gains[26], the system will tend to move back to its initial state of larger pumping lift with lower revenue loss (moving toward $P_C^4$ rather than $P_C^1$). This could be reasonable assuming that government aims for a sustainability-oriented policy, which may not be fully based on economic revenue. In addition to these effects, even when groundwater storage is recovered ($\Delta h$ reduces), the reduced groundwater pumping cost will create incentives for people to extract more groundwater (EP in Fig. 1b is shifted to the left in Fig. 1a), which again will eventually exacerbate groundwater depletion. This implies that once we are trapped in the situation with severe groundwater depletion, it will be difficult to move out of it. However, this negative effect can be potentially offset to some degree with higher penetration of SWE, which shifts $P_C^4 \to P_C^1$ down to $P_F^4 \to P_F^1$. Our results further demonstrate that if groundwater use is not regulated and the depletion keeps getting worse (pumping lift increases), then the benefits of higher SWE penetration is limited as the distance between the two wedge-shaped surfaces ($P_C^1 P_C^2 P_C^3 P_C^4$ and $P_F^1 P_F^2 P_F^3 P_F^4$) decreases. This suggests that the combined effect of more stringent control on groundwater abstraction plus SWE penetration is key to ameliorate revenue loss as well as benefit groundwater recovery. In summary, the results indicate that groundwater depletion can potentially diminish the added sustainable outcomes of SWE and we cannot merely assume that the positive effect of solar and wind penetration will persist indefinitely. Policy makers therefore have to take the long-term outlook of groundwater depletion into consideration when planning further deployment of SWE.

## Discussion

The recent severe and long-lasting drought in California triggered reforms to California's water policies in the short term to restrict water use (e.g., restrictions on urban water use). It also elevated an ongoing debate on future water policy changes to cope with such extreme events, such as establishing a groundwater banking market, banning water-intensive crops (e.g., almonds) and implementing quota-based water rights for efficient water allocation[27]. During the drought, the fast deployment of SWE helped compensate for the electricity deficit caused by the reduction of hydropower generation. Previously, SWE has only been recognized to facilitate air pollution mitigation and carbon emission reductions. Using the TF method and EP, this study provides new insights into the under-appreciated social value of SWE from the perspective of food security and environmental sustainability. The theoretical framework we have proposed can inform decision makers to design policies that can shift optimal water allocation

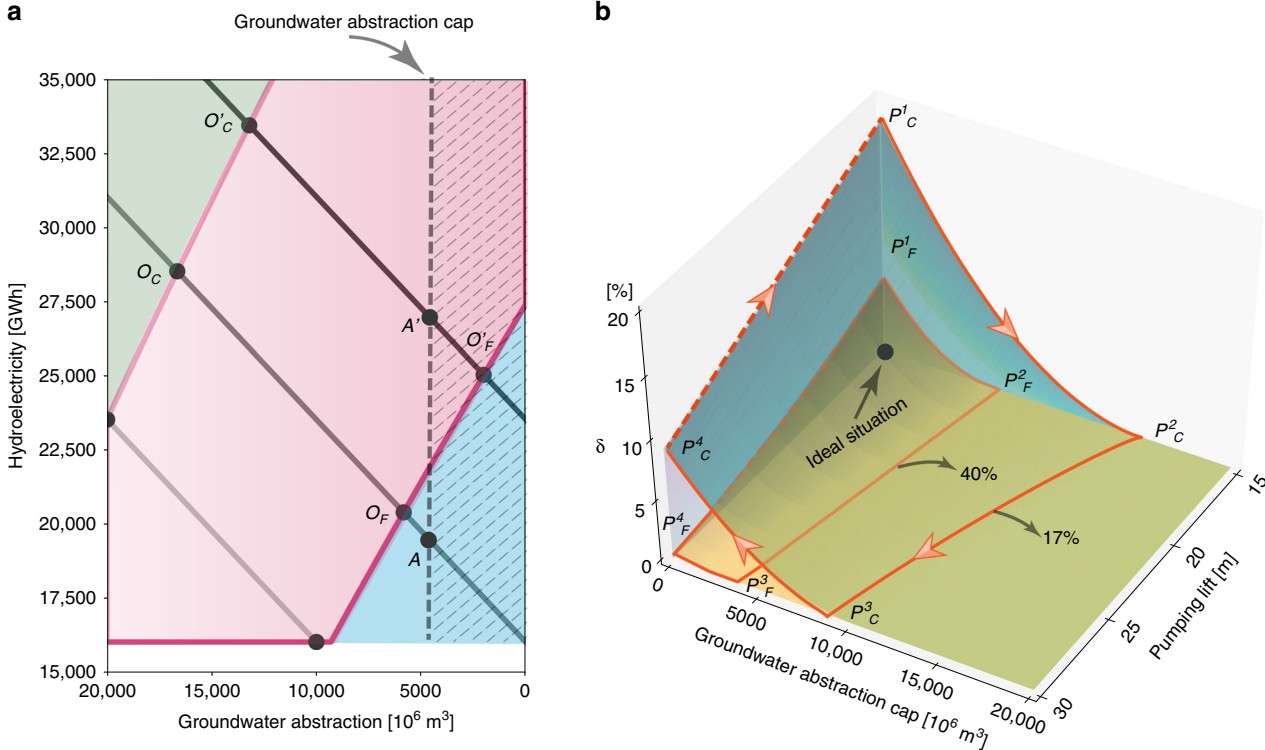

**Fig. 2** Impact of regulation policy on groundwater–hydropower trade-offs. **a** Schematic illustration of how groundwater abstraction cap ($g_w^{Cap}$) shifts the trade-offs optimal point. $O_C/O'_C$ and $O_F/O'_F$ represent the optimal point given current (17%) and future (40%) penetration of SWE in the normal/wet year without $g_w^{Cap}$. The vertical dashed line sets the limit of groundwater abstraction to meet certain regulations. With such a water constraint, the optimal point can only fall into the hatched area. When surface water is not abundant (e.g., during a normal year), the optimal point $O_C/O_F$ will not be attainable, and therefore $A$ becomes the new optimal under the regulation. However, such regulation does not affect the optimal point ($O'_F$) when water is abundant (e.g., during wet year) and when SWE penetration is high, as $O'_F$ is still in the hatched area. **b** Relative revenue loss ($\delta$) as a function of groundwater pumping lift ($\Delta h$) and $g_w^{Cap}$ under the influence of different penetration ratios of SWE (17% and 40%). Revenue loss zones are represented by the wedge-shaped area enclosed by the orange lines (current: $P_C^1 P_C^2 P_C^3 P_C^4$; future: $P_F^1 P_F^2 P_F^3 P_F^4$). Dashed orange lines represent extremely strict regulation policy (e.g., zero allowance of groundwater abstraction), which is unlikely to occur in reality. The black dot represents the ideal situation, where groundwater could be recovered and revenue loss is reduced

toward the target of ensuring food security with the aid of SWE. Furthermore, the high penetration of SWE may have additional value to increasing the society's resilience to drought given the following considerations. Electricity generated from SWE is intermittent and can be curtailed when it destabilizes power grids or becomes too abundant, especially as penetration level increases. Energy storage is therefore needed to enable excess electricity to be used and reduces the impact of SWE intermittency to the grid, enabling high penetration of SWE[28,29]. The deployment of energy storage provides co-benefits beyond the operation of the electricity grid. By using electricity stored from peak SWE generation, power systems would reduce reliance on hydropower, making both power generation and food production more resilient to drought. Furthermore, drought-tolerant SWE is substitutable for hydropower: less rainfall during a drought is associated with clearer skies and increased solar power generation. For example, state-wide solar power generation in California increased by 27% during the driest winter from November 2011 to March 2012 compared to the average generation in previous years (according to Clean Power Research).

The recent severe drought in California has acted as a catalyst to regulate unconstrained groundwater use, which has chronically lagged behind surface water regulations. In our study, we examine one possible groundwater regulation, which is to impose limits on groundwater abstraction, and analyze how it potentially feeds back to the social (i.e., water sustainability) value of penetrating SWE. We find that a groundwater abstraction cap could potentially reduce

crop irrigation and cause revenue loss, despite its benefit for environmental sustainability. Nevertheless, our results show that more stringent groundwater limitation (lower cap) would render less relative loss as SWE penetrates, and that SWE would largely alleviate the relative loss due to the increase of pumping lift. These findings highlight the co-benefits between the energy and environment in the sense that maintaining environmental (e.g., groundwater) sustainability can partially offset the impact of groundwater regulations on revenue loss. This is of critical importance for long-term policy making as we should not wait until groundwater further depletes to penetrate SWE. Otherwise, the added value of SWE in the future to balance the hydropower–groundwater trade-offs would be largely diminished. Our results suggest that it is beneficial to simultaneously deploy SWE and impose regulations on groundwater use earlier rather than later, since these two policies, when working together, facilitate each other to provide a greater combined benefit than either individual policy.

Although the particulars of the WFE nexus in each state or country differ, there is potential to generalize the proposed framework to other regions outside of California, especially in regions vulnerable to climate change and facing significant WFE challenges. While we focus on trade-offs between hydropower and groundwater, the general concept of TF–EP framework could be extended to a spectrum of sectors (e.g., domestic, industrial) and scales (e.g., from local to regional to global, and from the hourly basis of the electricity market to the seasonal basis of reservoir operation to the yearly basis of various water rights

regimes). Despite this general applicability, we note that the trade-offs and associated sustainability discussed here depend on the strong substitutability of SWE to hydropower in California. Lack of such substitutability may reduce the efficacy of this framework to balance the trade-offs and manage resource sustainability. For instance, in developing regions such as much of sub-Saharan Africa, hydropower still has price advantages compared to other forms of renewables. Unless stringent environmental regulations are imposed, the optimal solution to manage food energy trade-offs may not be obtainable under such a weak substitutability. Nevertheless, our approach can be implemented to provide insights for decision-making processes. For instance, given the complexity of upstream–downstream relationships in river basins, policy could be focused on creating incentives for upstream hydropower plants to release water for downstream irrigation. Such incentives could be in the form of government subsidies to operators of upstream hydropower plants using taxes paid by farmers in the downstream area if they are assured of water supply for irrigation. Alternative policy instruments can be designed to better price surface water based on experiences from Spain[30] and the European Water Framework Directive (WFD) or implement market-based systems to manage groundwater resources based on lessons drawn from the Murray–Darling Basin in Australia or Edwards Aquifer in Texas[31], which can assist California water agencies to meet the mandates of SGMA.

This study reveals some challenges which deserve further consideration and can be turned into opportunities for future improvement. Firstly, human dimensions are simplified in our framework, where we assume unchanging human behavior and decision making (e.g., water use, irrigation activities, crop choices). This assumption, which may be reasonable for the current focus, would need to be revisited in future work to better consider the co-evolution of the coupled human–natural system, as more reliable and consistent behavioral datasets (e.g., interviews, surveys) become available. Nevertheless, the proposed framework can be adjusted to integrate other factors into the optimization framework by including additional constraints. Considering improved technologies as an example, while future irrigation efficiency is expected to increase by 17–22% in California[32], this only has limited effect on the trade-off pathways under current penetration of SWE when water is limited (see Supplementary Fig. 1). With higher penetration of SWE, this limited effect disappears regardless of water availability conditions, highlighting the dominant role of SWE compared to improved irrigation efficiency. Although our TF–EP framework does not explicitly include the temporal dimension, we show how trade-off pathways vary with water availability, whose temporal variability is represented through hydrological simulations. Explicit consideration of how trade-offs vary over time (e.g., short-term reservoir operation versus long-term infrastructure investment) will be the next step, as this will enable a more flexible water management portfolio. This requires further work to extend current static framework to a dynamic version in the temporal space, and consider how future uncertainties of both water supply and demand will unfold as a result of climate change and human interventions. Addressing this is challenging but can be tackled through robust decision making (RDM) approaches combined with model-based large ensembles driven by different climate and social–economic scenarios[33]. Furthermore, this study focuses on the annual timescale without considering the intermittency of SWE. On short time scales (e.g., diurnal), the reduction of hydropower during a severe drought may result in a deficit between power supply and demand especially during peak demand hours and therefore jeopardize grid stability. To meet demand and cover shortfalls, either backup power, such as from natural gas, needs to be ramped up or additional electricity needs to be imported from neighboring grids. This is vital for regions whose baseload power source is hydropower, where additional regulation policies are required to control price volatility enhanced by large reductions of hydropower generation due to drought. In addition, the production function of hydroelectricity depends on the output from hydrological models, which work well at coarse scale resolution but may not capture the small-scale variability, especially for small hydropower plants. Besides, the spatial structure of California's water and energy grids are not considered, partly because our focus is on the state level as a whole. Including the spatial details of water diversions, reservoir operations, conjunctive water use, and energy transfer will increase the accuracy of the proposed framework, but it is also challenging because of the large uncertainties regarding current management rules of California's distributed water and energy infrastructure. More challenging is that these management activities are likely to adapt to a changing environment, but we cannot reasonably forecast them at this time. Moreover, the RPS target is only known at the state level and it is currently unclear how the target will be implemented at local scale (e.g., by utilities or Community Choice Aggregations). Therefore, the modeling framework represents how California as a whole would respond to future increases of renewables. Policy recommendations should be viewed with caution if this framework is to be applied at smaller scales, especially over regions where water markets and water rights play a dominant role in water allocation and its economic benefits and costs. Related to this, the current economic analysis could be extended beyond the private cost (i.e., pumping cost), to incorporate the social welfare (i.e., marginal opportunity cost associated with reducing future stock of the depletable nature of groundwater aquifer) by calculating the present value of current and future revenues of groundwater uses based on the Hotelling model[34]. It should be acknowledged that enhanced groundwater storage will also improve the reliability of local water supply for other non-irrigation uses and reduce the cost from other alternative water supplies (e.g., water transfer through canals and aqueducts), especially during drought years. Such added value due to increased supply reliability should be assigned economic value in future developments of the TF–EP framework.

In this study, we find that combining SWE within hydropower systems may achieve an under-appreciated mutual benefit for the WFE nexus. Here, by quantifying the water sustainability value of SWE using a TF framework, our case study in California has revealed the ability of increased SWE penetration to enhance drought resilience and groundwater sustainability. The resulting co-benefit on groundwater sustainability could further relieve the impact of groundwater regulations on agricultural revenue loss. However, SWE is no panacea for solving the WFE trilemma. Differences in RPS policies (in the USA), energy sources, hydro-climate variability, upstream–downstream relationships, and political and social constraints are likely to increase the complexity of rigorously managing the WFE nexus further than the archetype in this study. Complexity of the real world brings additional challenges to the scientific community in terms of how to incorporate the trade-off framework into large-scale hydrological or hydro-economic models. In models that these trade-offs are not considered, such exclusions can potentially influence the robustness of policies and the adaptability of the society to the changing environment. In this regard, previous work detailed in ref. [35] and its recent application to Australian land-sector sustainability[36] is an example of how this might be achieved. However, downscaling the trade-offs from large scales (e.g., state level) to small scales (e.g., county scale or grid scale) requires further investigation into complex topological networks of renewable power plants and their relationship with other sectors, which can be potentially simplified using computable general equilibrium (CGE) models.

## Methods

**Trade-off frontier (TF) and expansion path (EP).** The TF shows all efficient combinations of hydropower generation and groundwater abstraction given a certain surface water constraint. Construction of TF involves the following three steps. First, given groundwater abstraction, estimate how much surface water is required to meet total irrigation water demand. Second, given the remaining surface water, calculate how much hydropower can be generated based on an empirical production function between annual runoff and annual hydroelectricity[23]. Third, repeat steps 1 and 2 for all possible groundwater abstraction. Different from TF, the EP describes the combinations of groundwater abstraction and hydropower generation that the system will choose to maximize total revenue at each water constraint level. Mathematically, the TF and EP can be solved within an optimization framework. The objective is to maximize the total revenue ($R$) by solving the following non-linear optimization problem under a set of linear water constraints:

$$\underset{s_w^{\text{Hydro}}, s_w^{\text{Crop}}, g_w}{\text{maximize}} \quad R(s_w^{\text{Hydro}}, s_w^{\text{Crop}}, g_w) = B^{\text{Hydro}}(s_w^{\text{Hydro}}) - C^{\text{Pump}}(g_w) - D^{\text{Crop}}(s_w^{\text{Crop}}, g_w)$$

$$\text{subject to} \quad s_w^{\text{Hydro}} + s_w^{\text{Crop}} = s_w$$
$$s_w^{\text{Crop}} + g_w = \text{IWR}$$
$$0 \le s_w^{\text{Hydro}} \le s_w$$
$$0 \le s_w^{\text{Crop}} \le s_w$$
$$0 \le g_w \le \min\{g_w^{\text{Cap}}, \text{IWR} - s_w^{\text{Crop}}\}$$

$$(1)$$

where $B^{\text{Hydro}}$ [\$] is the economic profits from hydroelectric generation, $C^{\text{Pump}}$ [\$] is groundwater pumping cost, $D^{\text{Crop}}$ [\$] is the damage due to crop failure, $s_w^{\text{Hydro}}$ [$m^3$] is the surface water allocated for hydroelectric generation, $s_w^{\text{Crop}}$ [$m^3$] is the surface water allocated for crop irrigation, $s_w$ [$m^3$] is the total available surface water that can be allocated between hydropower production and irrigation, $g_w$ [$m^3$] is groundwater withdraw for crop irrigation, $g_w^{\text{Cap}}$ [$m^3$] is the groundwater cap due to regulations and IWR [$m^3$] is the irrigation water requirement (see next section for details) calculated from the hydrological model. Details on variables, parameters and numerical algorithms can be found in Supplementary Notes 1–6.

**Hydrological and water resources model.** Irrigation water requirement (IWR) is simulated with the Community Water Model (CWatM[37]), which is a macro-scale hydrological and water resources model developed by the Water Program at the International Institute of Applied Systems Analysis (IIASA). In this study, CWatM was forced by the daily meteorological forcing dataset WFDEI (WATCH Forcing Data methodology applied to ERA-Interim data[38]) at a 0.5° spatial resolution and daily temporal resolution covering the 34-year simulation period (1979–2012) (meteorological forcings and key parameters are described in Supplementary Note 2). CWatM inherits the same irrigation scheme as implemented in PCR-GLOBWB[8,39], which can separately estimate IWR for paddy and nonpaddy crops classified from the original 26 crop types in MIRCA2000 (ref. [40]). The irrigation scheme dynamically links the daily surface and soil water balance with irrigation water, which is more realistic compared to the existing irrigation schemes used in other large-scale hydrological models[8]. Details on the calculation of irrigation water for paddy and nonpaddy crops are provided in the supplementary materials (Supplementary Note 3).

We calibrated and validated CWatM against streamflow observations from eight USGS stations in California. Model calibration is performed using an evolutionary computational framework called Distributed Evolutionary Algorithms in Python[41] (DEAP). The modified version of the Kling–Gupta Efficiency (KGE, see equations in ref. [42]) is used as the objective function to be maximized. We have used a population size of 256 and recombination pool size of 32 with the number of generations set to 30 to calibrate CWatM, which proves to be sufficient to achieve convergence. Specifically, we have calibrated the model focusing on snow, evapotranspiration, soil, groundwater, routing process, lakes and reservoirs. Besides the correlation coefficient ($R$) and KGE, we also use percent bias ($B^{43}$) and Nash–Sutcliffe coefficient of efficiency (NSE[44]) to evaluate the performance of CWatM. Time series of observed and simulated streamflow and the associated performance metrics for the calibration and validation periods can be found in the supplementary materials (Supplementary Figs. 2–17). Results demonstrate that CWatM can well reproduce the streamflow variability and magnitude both at daily and monthly time scale.

**Reporting summary.** Further information on research design is available in the Nature Research Reporting Summary linked to this article.

## Data availability

Hydrological simulations from CWatM are available upon request to X.H. Hydroelectricity production can be obtained from the U.S. Energy Information Agency (https://www.eia.gov/). Revenue of field crops can be obtained from Figure 4 in ref. [45].

## Code availability

CWatM codes can be obtained from IIASA's Water Program at http://www.iiasa.ac.at/cwatm and https://cwatm.github.io. Pseudocode to calculate the optimal point and expansion path can be found in the supplementary material (Supplementary Notes 4–6). Matlab codes for trade-off analysis and Python plotting scripts are available upon request to X.H.

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

## Acknowledgements

X.H. would like to thank Prof. Michael Oppenheimer at Princeton University, Prof. Jim Hall at Oxford University, and Dr. Declan Conway and Dr. Kate Gannon at the London School of Economics for helpful discussions and comments. Part of the research was developed in the Young Scientists Summer Program at the International Institute for Applied Systems Analysis, Laxenburg (Austria), with financial support from the USA National Member Organization. This material was also based upon work supported by NOAA grant NA14OAR4310218 and the Science, Technology, and Environmental Policy (STEP) fellowship by the Princeton Environmental Institute at Princeton University. J.S. is supported through the UK Research and Innovation Global Challenges Research Fund projects "Building REsearch Capacity for sustainable water and food security In drylands of sub-saharan Africa (BRECcIA)", grant number NE/P021093/1 and "FutureDAMS: Design and Assessment of resilient and sustainable interventions in water-energy-food-environment Mega-Systems", grant number ES/P011373/1.

## Author contributions

X.H. and J.S. conceived the research and drafted the manuscript. X.H. and K.F. performed the trade-off analysis. X.H. performed the hydrological simulation with help from P.B. X.H. designed and prepared the figures. X.L., A.B.C., Y.W., P.B. and E.F.W. provided comments.

## Competing interests

The authors declare no competing interests.
