## [Peer Review File · Nature Communications]

Reviewers' comments:

Reviewer #1 (Remarks to the Author):

This is an interesting paper, offering a simple analysis of hydropower-irrigation tradeoffs with groundwater as an alternate to surface water for irrigation and solar and wind energy (SWE) as a substitute for hydropower. The underlying idea is conceptually straightforward. Given that water can be used either for irrigation or for hydropower, if SWE can offset the use of stored surface water for power generation, then more can be made available for irrigation and thus less groundwater will need to be pumped. The paper offers a quantitative assessment of these tradeoffs for California.

I think some discussion about the basic premise is warranted. First, last year a bill was passed requiring that 50 percent of California's electricity be powered by renewable resources by 2025 and 60 percent by 2030. These are not easy targets and using one renewable source (SWE) to reduce another (hydropower) would make it even harder to hit them. What does this say about the feasibility of the tradeoff analysis presented? Second (somewhat related), assuming that SWE penetration directly reduces the demand for hydroelectric power means that SWE is viewed as completely additional capacity not replacing another source. Is this reasonable? What is special about SWE for the tradeoff analysis? Any alternate source of electricity that could reduce hydropower demands would have the same effect, wouldn't it?

I am perplexed by the discussion about how it is difficult to recover groundwater storage once it has been depleted. PC4 -> PC1 is for zero groundwater abstraction (mandated), presumably indicating that PC1 is the steady-state level for a "recovered" system. The argument then invokes loss aversion to claim that irrigators would resume pumping because PC4 has a lower delta. But the mandate is for zero abstraction! I presume that my confusion is just a reflection of how the paragraph is worded. Even so, it seems to me that most of the paper uses a rational actor framework for maximizing revenue and suddenly a switch is made to minimizing revenue loss. Here again I am a bit confused. Prospect theory (Kahneman and Tversky) is an asymmetrical approach to maximizing utility, not to minimizing losses so I assume that the authors intend something that I am not understanding.

I think that the authors may reach a bit far in some of the discussion of the implications of their results. "By using electricity stored from peak SWE generation, power systems would reduce reliance on hydropower." One of the significant advantages of hydropower as a renewable is that it is dispatchable and one of the most readily available methods to store electricity from peak SWE generation would be pumped storage. But water for irrigation is a consumptive use. Thus, the authors' claim can be read to be that large-scale battery storage or some other technology not yet fully developed will be ready to deploy. This seems overly optimistic to me. The authors acknowledge some of the complexities toward the end of the discussion but nevertheless retain some rather strong statements about the utility of the framework, for example about how the UN SDG's can be met.

Overall I think the paper represents a solid contribution. As noted above, some clarifications and possibly toning down some of the discussion would improve the paper in my opinion.

Reviewer #2 (Remarks to the Author):

This paper covers the interesting topic of the food-energy-water nexus in California and its optimization to inform groundwater policy, and in particular to evaluate the increased use of renewable energy sources (pv, wind). The reviewer has identified the following issues:

It wasn't clear what the spatial structure of the analysis is. A calibrated hydrological model at several stations is provided, but it isn't clear how this is connected to California's distributed water and energy system. For example what is the spatial unit of analysis for groundwater? The paper describes using groundwater parameter data from Knapp et al. 2003, but that study was just for

Kern county, i.e., a relatively small region within California. California's Central Valley has several distinct aquifers some unconfined other confined; are these represented in this groundwater themed analysis?

Are urban water demands represented? And if so at what spatial scale, and how are these demands connected to California's water supply network?

California's water management uses infrastructure – both for transmission (transfers) and storage (reservoirs). How water is managed and allocated via this infrastructure determines the benefits obtained from water by the different sectors. Also, how water is stored and allocated in different times of the year, via water rights and water markets, determines the economic benefits from water use. Particularly under scarcity water markets become active and determine the economic costs of scarcity. Finally, in California groundwater and surface water are used conjunctively in "water banks". Is the reviewer correct that these essential aspects of California's water management have been ignored in this study? If so, is there a danger of making policy recommendations about a hydrologic-engineered-economic system without considering with sufficient accuracy how water supply and demand manifest over space and time.

The optimization is monetary; does that imply environmental, engineering and social aspects are not considered? For example, how is supply reliability monetized in this framework?

The authors have applied a form of annual dynamic optimization but results are not given over time, so it is not possible for readers to evaluate how the model represents the system's evolution over time. Does water use change over time? Do water users learn or evolve? Do farmers make decisions about crop choice or irrigation technologies? The paper speaks of finding an optimal pathway (over time the reviewer assumes) yet recent literature on development pathways underscores the importance of considering the uncertainty of both future supply and demand which doesn't seem to be considered in this analysis. Does the annual dynamic economic optimization only consider one scenario, the historical one?

Trade-offs between economic, engineering and social benefits are relevant when managing nexus systems, yet this paper reports on efficient combinations of hydropower generation and groundwater use. The paper describes a framework for nexus systems and sustainable development goals (SDGs) but it isn't clear how groundwater use constitutes an SDG and whether trade-offs between economic production (hydropower) and individual supply sources (groundwater in this paper) are relevant to policy makers.

Reviewer #3 (Remarks to the Author):

The submitted manuscript proposes a trade-off frontier framework to account for the role of solar and wind energy in the water-food-energy nexus. The idea is interesting, innovative and promising, but the manuscript fails in presenting it in a clear and intelligible way. A lot of confusion arises when describing the trade-off frontier (TF) – actually there is no explanation supporting the construction of the TF curve. Also, both figures are extremely dense, the caption is poor as well as their description in the main text.

I strongly encourage the authors to significantly re-define the manuscript structure and better clarify their main messages and outcomes. As it stands, I cannot recommend this paper to be published in Nature Communications its present form.

Some comments:

p. 6 l. 98: authors should explain how water surface constrains vary from dry to wet in Fig 1

p. 6 l. 104: where "inefficient/unattainable" status could be identified in TF? Please add an explanation

p.7 ll. 116-118: The authors state that the black point in Fig 1 represents the condition such that maximum revenue and efficient water allocation are met. Why maximum revenue? The iso-revenue line shows equal revenues for different groundwater and hydropower conditions... I found

this part quite obscure

p. 7 l. 123: inwards and outwards: what is the reference?

p. 8 ll. 142-143: why do the current scenario shows a smaller curvature of iso-revenue lines compared to future scenario? Also, what is the % of hydropower production now? And in 2030?

p. 20 eq 1: is there an error in the 3rd water constraint (maybe a sum instead of a product)?

Figs S1-S8: a detailed explanation of plots and tables should be added

2. Responses to Reviewer #1

This is an interesting paper, offering a simple analysis of hydropower-irrigation tradeoffs with groundwater as an alternate to surface water for irrigation and solar and wind energy (SWE) as a substitute for hydropower. The underlying idea is conceptually straightforward. Given that water can be used either for irrigation or for hydropower, if SWE can offset the use of stored surface water for power generation, then more can be made available for irrigation and thus less groundwater will need to be pumped. The paper offers a quantitative assessment of these tradeoffs for California.

Response: Thanks for the positive comments.

I think some discussion about the basic premise is warranted. First, last year a bill was passed requiring that 50 percent of California's electricity be powered by renewable resources by 2025 and 60 percent by 2030. These are not easy targets and using one renewable source (SWE) to reduce another (hydropower) would make it even harder to hit them. What does this say about the feasibility of the tradeoff analysis presented? Second (somewhat related), assuming that SWE penetration directly reduces the demand for hydroelectric power means that SWE is viewed as completely additional capacity not replacing another source. Is this reasonable? What is special about SWE for the tradeoff analysis? Any alternate source of electricity that could reduce hydropower demands would have the same effect, wouldn't it?

Response: We want to emphasize that our analysis focuses on time period of drought (which will likely get worse in the future), during which trade-offs exist and conflicts between hydropower and irrigation are exacerbated. When water is abundant, there are no such trade-offs, and penetration of SWE will not necessarily reduce hydropower. Therefore, reducing electricity generation from large-scale hydropower will not influence the compliance of RPS directly. However, as a baseload resource, large-scale hydro provides resource adequacy and facilitates the integration of solar and wind.

To be clear, in our framework SWE does displace natural gas to complement the peak demand. This is how we estimate the shadow price of hydropower as detailed in the Supplementary material S1 (see Equation 2) based on the natural gas price.

SWE does not have any special role in the trade-off analysis. We chose SWE simply because California has high renewable portfolio standards (RPS), which mandate the state to have large penetration of SWE. Meanwhile, their unrecognized value related to sustainable water use has not been examined in previous studies. In our trade-off framework, penetration of SWE is exogenous, which means it could be replaced by other policy instruments, depending on the specific policy question. For example, if some regions are developing nuclear energy, then this could potentially have similar effects as SWE on reducing the reliance on hydropower during drought periods.

I am perplexed by the discussion about how it is difficult to recover groundwater storage once

it has been depleted. $PC_4 \rightarrow PC_1$ is for zero groundwater abstraction (mandated), presumably indicating that PC_1 is the steady-state level for a “recovered” system. The argument then invokes loss aversion to claim that irrigators would resume pumping because PC_4 has a lower delta. But the mandate is for zero abstraction! I presume that my confusion is just a reflection of how the paragraph is worded. Even so, it seems to me that most of the paper uses a rational actor framework for maximizing revenue and suddenly a switch is made to minimizing revenue loss. Here again I am a bit confused. Prospect theory (Kahneman and Tversky) is an asymmetrical approach to maximizing utility, not to minimizing losses so I assume that the authors intend something that I am not understanding.

Response: We apologize for such confusion because of the wording. We have revised our manuscript to clarify our argument in Lines 213-223 on Pages 12-13: “This is because a small reduction of pumping lift (slightly recovery of groundwater storage) would result in a significant increase in revenue loss (i.e., δ increases dramatically along the direction of $PC^A \rightarrow PC^I$ with a slight decrease of Δh). As δ is defined as a relative term, it shows how much total revenue will be reduced if groundwater regulation policy is added as a constraint relative to the situation without such constraint. Given that people tend to be loss averse (e.g., minimize relative revenue losses compared to the optimal situation) and prefer to make decisions based on losses rather than gains (Kahneman and Tversky, 1979), the system will tend to move back to its initial state of larger pumping lift with lower revenue loss (moving towards PC^A rather than PC^I). This could be reasonable assuming that government aims for a sustainability-oriented policy, which may not be fully based on economic revenue.”

The reviewer is right that our optimization is based on a rational actor framework for maximizing revenue. However, the reviewer may have misunderstood our discussion on revenue loss. According to our definition (Lines 190-192 on Page 11), the revenue loss is a relative term. It shows how much total revenue will be reduced if groundwater regulation policy is added as a constraint relative to the situation without such constraint.

The rational trade-off framework for farmers and power plants does not conflict with the guiding ideology for policy design of local government, which is not searching for the temporal rational optimal, but keeping a balance between current revenue and long-term sustainability. Specifically, we assume that the government would like to maximize the total revenue for farmers and hydropower plants, but at the same time, wants to design an appropriate groundwater cap to ensure that the revenue loss is as small as possible compared to the best scenario (which can get maximum profit without constraint from groundwater regulation). This can be achieved by loosening the regulation (i.e., increase the cap), but this will create incentives for farmers to continue pumping, which will lead to groundwater depletion. This is reflected in our statement: “once we are trapped in the situation with severe groundwater depletion, it will be difficult to move out of it.”

I think that the authors may reach a bit far in some of the discussion of the implications of their results. “By using electricity stored from peak SWE generation, power systems would reduce reliance on hydropower.” One of the significant advantages of hydropower as a renewable is

that it is dispatchable and one of the most readily available methods to store electricity from peak SWE generation would be pumped storage. But water for irrigation is a consumptive use. Thus, the authors' claim can be read to be that large-scale battery storage or some other technology not yet fully developed will be ready to deploy. This seems overly optimistic to me. The authors acknowledge some of the complexities toward the end of the discussion but nevertheless retain some rather strong statements about the utility of the framework, for example about how the UN SDG's can be met.

Response: We agree with the reviewer that pumped storage is one of the most readily available methods to store electricity from peak SWE generation, but good locations in California for pumped storage are already developed and would require very high upfront capital costs to develop new projects (CEC, 2016, Bulk Energy Storage in California). The other concern is that pumped storage projects have long construction time (e.g., 10 years), due to securing permits, building reservoirs and tunnels, as well as resistance because of the potential environmental impacts.

The technology of large-scale battery storage is ready. Although battery storage is not cost competitive in the current market, various projections show steep decline of costs in the next decade or so. Short-duration battery storage is being picked in resource planning by utilities. For example, recently, Southern California Edison, the second largest electricity utility in California, chose to build and own a 100/400 MW-hour battery system instead of the 262 MW natural gas peaker plant it previously had chosen (Spector, 2019). Although this is just a local case, in the long term as long-duration battery costs come down, California will rely on battery storage to facilitate solar and wind integration not only for its capacity value but also energy value. There could be a variety of approaches, not only from large-scale batteries and pumped storage, but also from other relatively cheaper technologies including compressed air, geothermal carbon capture and storage.

As for UN SDG, our intention was to note that the framework could be useful in other contexts including analysis of trade-offs between SDGs, which is vital to understand because of how intertwined the goals are. Nevertheless, we agree that our statements may be premature, and so we have deleted them in the abstracts as well as the discussion section to tone down the conclusion.

References:

Spector, 2019, Southern California Edison Picks 195MW Battery Portfolio in Place of Puente Gas Plant. Greentech Media.

CEC, 2016, Bulk Energy Storage in California. Available:

<https://ww2.energy.ca.gov/2016publications/CEC-200-2016-006/CEC-200-2016-006.pdf>

Overall I think the paper represents a solid contribution. As noted above, some clarifications and possibly toning down some of the discussion would improve the paper in my opinion.

Response: Thanks again for the positive and constructive comments. As explained above, we have clarified our main findings and toned down some of our conclusions.

2. Responses to Reviewer #2

This paper covers the interesting topic of the food-energy-water nexus in California and its optimization to inform groundwater policy, and in particular to evaluate the increased use of renewable energy sources (pv, wind). The reviewer has identified the following issues:

Response: Thanks for the constructive comments. Below is the point-by-point response.

It wasn't clear what the spatial structure of the analysis is. A calibrated hydrological model at several stations is provided, but it isn't clear how this is connected to California's distributed water and energy system. For example what is the spatial unit of analysis for groundwater? The paper describes using groundwater parameter data from Knapp et al. 2003, but that study was just for Kern county, i.e., a relatively small region within California. California's Central Valley has several distinct aquifers some unconfined other confined; are these represented in this groundwater themed analysis?

Response: Thanks for the valuable comments. Our trade-off analysis was conducted over the entire California (this is the spatial unit of analysis of groundwater) and we did not consider the complex spatial structures because of the following reasons:

(1) The focus of our paper is to develop a conceptual trade-off framework that is general enough to be transferable to other regions and other situations. Therefore, the paper itself is a trade-off in the sense that we need to consider the balance between model complexity and model generalizability. For this purpose, we need to make certain assumptions and simplify our model structures. This is part of the reason why we treat California as a whole.

(2) Certain hydrological variables (e.g., irrigation water requirement, surface water availability) are required as constraints for the hydro-economic optimization model. The spatial variability of these variables is incorporated in the hydrological model (CWatM), which has been calibrated with local data. These variables are firstly estimated at pixel level and then we aggregate them to the state level for the trade-off analysis. However, it is difficult to incorporate the spatial structure of California's water and energy grids into CWatM. Including the spatial details (such as water diversions, reservoir operations, conjunctive water use, and energy transfer) will increase the accuracy of the proposed framework, but it is also challenging because of the large uncertainties regarding current management rules of California's distributed water and energy infrastructure. More challenging is that these management activities are likely to adapt to a changing environment, but we cannot reasonably forecast them at this time.

(3) Furthermore, the renewable energy policy target is designed for the entire CA, and at smaller scales, future penetration of SWE is generally unknown or is difficult to estimate in a reliable and consistent way.

(4) We argue that our framework is flexible enough to be coupled with other high-resolution and California-tailored hydrological models, such as C2VSim (developed by California Department of Water Resources) and CVHM (developed by USGS). But this will require a better understanding of the topology of the water grids as well as the energy grids, and how

they interact with each other. Again, if these models are used, this will inevitably limit their application over other regions outside of California and beyond.

We therefore acknowledge that our framework does not reflect the complexities of the CA water resources and energy production systems, but does take into the spatial variability in hydrology. We acknowledge the limitations and add some discussion on this from Lines 316-375 on pages 17-20: “This study reveals some challenges which deserve further consideration and can be turned into opportunities for future improvement. Firstly, human dimensions are simplified in our framework, where we assume unchanging human behavior and decision making (e.g., water use, irrigation activities, crop choices). This assumption, which may be reasonable for the current focus, would need to be revisited in future work to better consider the co-evolution of the coupled human-natural system, as more reliable and consistent behavioral datasets (e.g., interviews, surveys) become available. Nevertheless, the proposed framework is flexible enough to integrate other factors into the optimization framework, which can be done by including additional constraints. Considering improved technologies as an example, while future irrigation efficiency is expected to increase by 17-22% in California (Cooley et al., 2014), this only has limited effect on the trade-off pathways under current penetration of SWE when water is limited (see Figure S1). With higher penetration of SWE, this limited effect disappears regardless of water availability conditions, highlighting the dominant role of SWE compared to improved irrigation efficiency. Although our TF-EP framework does not explicitly include the temporal dimension, we show how trade-off pathways vary with water availability, which indeed vary over time. Further work is needed to extend this static framework to a dynamic version in the temporal space and consider future uncertainties of both water supply and demand due to changing climate and human interventions. Such challenges can be tackled through robust decision making (RDM) approaches combined with model-based large ensembles driven by different climate and social-economic scenarios (He et al., 2019). Further, this study focuses on the annual timescale without considering the intermittency of SWE. On short time scales (e.g., diurnal), the reduction of hydropower during a severe drought may result in a deficit between power supply and demand especially during peak demand hours and therefore jeopardize grid stability. To meet demand and cover shortfalls, either backup power, such as from natural gas, needs to be ramped up or additional electricity needs to be imported from neighboring grids. This is vital for regions whose baseload power source is hydropower, for example North Carolina, where additional regulation policies are required to control price volatility enhanced by large reductions of hydropower generation due to drought. In addition, the production function of hydroelectricity depends on the output from hydrological models, which work well at coarse scale resolution but may not capture the small-scale variability, especially for small hydropower plants. Besides, the spatial structure of California’s water and energy grids are not considered, partly because our focus is on the state level as a whole. Including the spatial details of water diversions, reservoir operations, conjunctive water use, and energy transfer will increase the accuracy of the proposed framework, but it is also challenging because of the large uncertainties regarding current management rules of California’s distributed water and energy infrastructure. More challenging is that these management activities are likely to adapt to a changing environment, but we cannot reasonably forecast them at this time. Moreover, the RPS target is only known

at the state level and it is currently unclear how the target will be implemented at local scale (e.g., by utilities or Community Choice Aggregations). Therefore, the modeling framework represents how California as a whole would respond to future increases of renewables. Policy recommendations should be viewed with caution if this framework is to be applied at smaller scales, especially over regions where water markets and water rights play a dominant role in water allocation and its economic benefits and costs. Related to this, the current economic analysis could be extended beyond the private cost (i.e., pumping cost), to incorporate the social welfare (i.e., marginal opportunity cost associated with reducing future stock of the depletable nature of groundwater aquifer) by calculating the present value of current and future revenues of groundwater uses based on the Hotelling model (Hotelling, 1931). It should be acknowledged that enhanced groundwater storage will also improve the reliability of local water supply for other non-irrigation uses and reduce the cost from other alternative water supplies (e.g., water transfer through canals and aqueducts), especially during drought years. Such added value due to increased supply reliability should be assigned economic value in future developments of the TF-EP framework.”

We agree that parameters from Kern county may not be representative for the entire California given that California has distinct aquifers. We have therefore recalculated the pumping cost (see Equation in Text S1) and redone the analysis using updated parameter values for s_y (specific yield) and c (unit pumping cost) (details are summarized in Table R1). Comparison between the original results and revised results can be found in Figure R2. We examine the sensitivity of the optimal path to the aquifer characteristics by varying s_y from 0.10 to 0.20 (Figure R3). As it is difficult to obtain reliable data for parameter k (cost of groundwater equipment use), sensitivity tests are conducted by varying k from 0.40 to 0.60 (ha cm)⁻¹ ($\pm 20\%$ of the original value obtained from the Kern study, see Figure R4).

Table R1. Comparison of parameter values used in the initial submission and revision.

Parameter	Initial submission		Revision	
	Value	Reference	Value	Reference
s_y	0.130	Knapp et al. (2013)	0.145 ⁽¹⁾	Brush et al. (2013)
c	0.039 /ha/cm/m	Knapp et al. (2013)	0.053 /ha/cm/m ⁽²⁾	Chou et al. (2012)

Note (1): Estimation of specific yield (s_y) is challenging. Most production aquifers in California’s Central Valley contain a mixture of sands, gravels, and finer sediments. According to C2VSim (Brush et al., 2013, Figure R1), typical s_y ranges from 0.10 to 0.20 over the Central Valley. Based on the s_y map, we calculate the area-weighted s_y over the entire Central Valley, whose value (0.145) is close to the values (0.13) used for Kern County.

Note (2): Based on Chou et al. (2012), the average groundwater pumping cost over California is \$0.20 per acre-feet per feet of lift in year 2000 dollars. This value is adjusted for inflation ($\times 1.346$) in our revised manuscript to calculate the cost related to equipment use.

Figure R1. Map of specific yield (from C2VSim hydrological model, Brush et al., 2013).

Overall, we find that the results do not change substantially with the updated parameter values of s_y and c (Figure R2). Sensitivity tests (Figure R3) show that expansion paths (EP) are sensitive to s_y , especially for scenarios with lower penetration of SWE (yellow lines). However, our general conclusions are still robust to such sensitivity, as there is still a clear separation between future and current EPs with future EPs always being to the right side of current EPs. In other words, regardless of s_y , higher penetration of SWE always reduces the groundwater abstraction and therefore increases groundwater sustainability. The same conclusion holds for the sensitivity analysis for k (Figure R4), where we also identify a clear shift of EP towards the right with higher penetration of SWE. The results are within our expectation, as higher k values mean higher equipment use, which makes groundwater more expensive and therefore leads to reduced groundwater abstraction. Although our general conclusion is not influenced by choosing different parameters, these sensitivity tests highlight the need to collect reliable and consistent datasets to appropriately parameterize s_y and k if the trade-off framework is to be applied in different regions.

We have included these sensitivity tests in the Supplementary material Figures S2-S3. A brief discussion of these sensitivity tests has been added in the supplementary material Text S1 on Pages 2-3: “Due to lack of reliable datasets, parameter values are either taken from the literature or derived from model input parameters. For instance, based on Chou et al. (2012), the average groundwater pumping cost, c , in California is \$0.20 per acre-foot per foot of lift in year 2000 dollars. We adjust this value for inflation to calculate the unit pumping cost. As for s_y , we calculate the area-weighted value based on the s_y map provided by the C2VSim hydrological model (Brush et al., 2013). Given the high spatial variability of s_y over California (Brush et al., 2013), sensitivity tests are performed to examine the robustness of optimal expansion paths (EP) to s_y (see Figure S2). As there are no reliable and consistent estimates for k across California,

we took the value from Table A1 in Knapp et al. (2003), but with a $\pm 20\%$ variation of the original value to account for parameter uncertainty (see Figure S3). Sensitivity tests (Figure S2 and S3) show that the actual expansion paths (EP) are sensitive to s_y and k . However, our general conclusions are still robust to such sensitivity, as there is a clear separation between future and current EPs with future EPs always being to the right side of current EPs. In other words, regardless of s_y , higher penetration of SWE always reduces the groundwater abstraction and therefore increases groundwater sustainability. Although our general conclusion is not influenced by choosing different parameters, these sensitivity tests highlight the need to collect reliable and consistent datasets to appropriately parameterize s_y and k if the trade-off framework is to be applied in different regions.”

Figure R2. Optimal expansion paths under current (yellow lines, 17%) and future (red lines, 40%) penetration of SWE using original parameters (from Knapp et al., 2003, dashed lines) and updated parameters (solid lines) with different groundwater pumping lift (Δh).

Figure R3. Sensitivity of expansion paths to specific yield s_y under current (yellow lines, 17%) and future (red lines, 40%) penetration of SWE with different groundwater pumping lift (Δh). Higher values of s_y shift the EP to the left as groundwater pumping costs are reduced, which leads to higher amount of groundwater abstraction. Here s_y ranges from 0.10 to 0.20 based on the C2VSim model (Brush et al., 2013).

Figure R4. Same as Figure R3 but showing the sensitivity of EP to the energy cost parameter k ranging from 0.40 to 0.60 (ha cm)⁻¹. With higher k values, EP moves rightward, because higher equipment cost makes groundwater more expensive and therefore leads to reduced groundwater abstraction.

References:

- Brush, C.F., Dogrul, E.C., and Kadir, T.N. June 2013. Development and Calibration of the California Central Valley Groundwater-Surface Water Simulation Model (C2VSim), Version 3.02-CG.
- Heidi Chou. 2012. Groundwater Overdraft in California's Central Valley: Updated CALVIN Modeling Using Recent CVHM and C2VSIM Representations. Master thesis. University of California, Davis.

Are urban water demands represented? And if so at what spatial scale, and how are these demands connected to California's water supply network?

Response: Yes, urban water demand is represented in our physical hydrological model (CWatM) at the pixel level. Demand is linked to population densities, which are connected to the urban water supply network. In our coupled hydro-economic optimization framework, we assume that pixel-level urban water demands are satisfied first. The remaining available surface water at the pixel level is then aggregated to the whole of California. This aggregated total surface water availability is then used as the constraint in the optimization model.

Urban water demands are connected to California's water supply network in the CWatM model through local and upstream reservoirs. However, major water transfer projects in California

have not been included in the model due to the unavailable conveyance capacity data and the complex operational and regulatory rules, as well as the limitations of the model to represent such transfers. However, this has been partly compensated as our study focuses California as one hydro-economic unit.

California's water management uses infrastructure – both for transmission (transfers) and storage (reservoirs). How water is managed and allocated via this infrastructure determines the benefits obtained from water by the different sectors. Also, how water is stored and allocated in different times of the year, via water rights and water markets, determines the economic benefits from water use. Particularly under scarcity water markets become active and determine the economic costs of scarcity. Finally, in California groundwater and surface water are used conjunctively in “water banks”. Is the reviewer correct that these essential aspects of California's water management have been ignored in this study? If so, is there a danger of making policy recommendations about a hydrologic-engineered-economic system without considering with sufficient accuracy how water supply and demand manifest over space and time.

Response: The reviewer is correct. Again, as mentioned earlier in our response, we sought to find a balance between model complexity and model generalizability. California has one of the world's most complicated water management systems because of the infrastructure, water markets, water rights and institutions. It is not possible to incorporate such complexity in our current large-scale hydrological model. Instead, the focus of our paper is to develop a simple and transparent conceptual framework to investigate the aggregated behavior of hydropower-irrigation trade-offs *at the state level* rather than examining small-scale interactions across space. In this regard, we think the trade-off framework can still be useful at the large scale (i.e., state level) to provide policy insights. We agree with the reviewer that our current framework has limitations if applied over space and therefore, policy recommendations at smaller scales (e.g., sub-region of California) should be avoided. We added some discussion on the limitations of our framework in Lines 349-363 on pages 18-19.

The optimization is monetary; does that imply environmental, engineering and social aspects are not considered? For example, how is supply reliability monetized in this framework?

Response: Environmental aspects are considered in our physical model simulations. We incorporate environmental flow requirements in CWatM to ensure sufficient water for environmental needs. Certain engineering aspects are also considered, such as reservoir operation, but only for major reservoirs in California. Again, due to current model limitations, major water transfer projects are not included in CWatM. Social aspects (e.g., water use behavior) are not included, but this could be done through the coupling of CWatM with agent-based modeling.

As explained in an earlier response, our focus is on the trade-off between hydropower and irrigation water use. Therefore, supply reliability is not monetarized in our framework. The other reason why we did not consider supply reliability is because its benefits and costs (e.g.,

avoided water supply costs from other alternatives, such as water transfer costs) are uncertain and difficult to estimate. However, we acknowledge that the value of additional supply reliability provided by increased groundwater storage could be significant into the future. Therefore, we added some qualitative discussion on this benefit in Lines 370-375 on Pages 19-20: “It should be acknowledged that enhanced groundwater storage will also improve the reliability of local water supply for other non-irrigation uses, and reduce the cost from alternative water supplies (e.g., water transfer through canals and aqueducts) especially during drought years. Such added value due to increased supply reliability should be assigned economic value in future developments of the TF-EP framework.”

The authors have applied a form of annual dynamic optimization but results are not given over time, so it is not possible for readers to evaluate how the model represents the system’s evolution over time. Does water use change over time? Do water users learn or evolve? Do farmers make decisions about crop choice or irrigation technologies? The paper speaks of finding an optimal pathway (over time the reviewer assumes) yet recent literature on development pathways underscores the importance of considering the uncertainty of both future supply and demand which doesn’t seem to be considered in this analysis. Does the annual dynamic economic optimization only consider one scenario, the historical one?

Response: In our initial submission, our annual optimization framework is demonstrated over three representative years (dry, normal, and wet) because the optimal pathway we defined is with respect to the changing water availability, not explicitly over time. The hydrological conditions for these three years are from the historical period. We appreciate the reviewer’s insights in terms of representing the system’s evolution over time and suggestions to consider future uncertainties of supply and demand. But this has been a focus for a separate study (He et al., 2019, *Identifying robust development pathways to manage the groundwater-food-energy trilemma in California through penetration of renewable energy generation*, AGU Fall Meeting Abstract) where we demonstrate that our framework can be easily applied over continuous time series to investigate system’s evolution. Just to briefly explain here. To account for future uncertainties of water supply from climate forcings, bias-corrected future projections from four different Global Climate Models (GCMs) (i.e., GFDL-ESM2M, IPSL-CM5A-LR, MIROC-ESM-CHEM, NorESM1-M) under RCP6.0 (Representative Concentration Pathway, medium emissions) are used to drive CWatM to estimate surface water availability and irrigation water requirement during 2016-2030. Projections of future water demand from other sectors (i.e., domestic, industrial, and livestock) are obtained from the Water Futures and Solutions (WFaS) Initiative (Wada et al., 2016) under the SSP2 (Shared Socioeconomic Pathways, middle-of-the-road scenario indicating a medium-level of adaptation and mitigation) scenario. As a proof-of-concept, here we only focus on the combination of RCP6.0 and SSP2 to approximate the middle-of-the-road projections for future climate and socioeconomic changes.

In this response letter, we have performed additional analysis to specifically address the following questions from the reviewer:

- (1) Optimization over time: Dynamic optimization from 2016 to 2030 is included.

(2) Changing water use: We agree with the reviewer that water use does change over time and water users can learn and evolve. This has been included in the dynamic optimization framework, and future water use data based on SSP2 scenario is obtained from WFaS.

(3) Uncertainties from water supply and demand: These are represented by using ensemble hydrological simulations driven by four different GCMs.

(4) Sensitivity of optimal pathways to changing irrigation efficiency: As it is difficult to incorporate farmers' decisions on crop choice due to lack of data, here we only consider potential changes of irrigation technology. Based on Cooley et al. (2014), future irrigation efficiency in California is expected to increase by 17-22% in all year types (i.e., normal, dry, wet). We conducted sensitivity analysis to examine how this will impact the optimal trade-off pathway.

Results considering (1)(2)(3) are presented in Figure R5. Results considering (4) are shown in Figure R6 (also included in the supplementary material as Figure S1).

Figure R5 shows the percentage change of future (2016-2030) projected recovered groundwater (Δ RGW, see the inset schematic diagram in Figure R5) due to higher penetration of SWE. This temporal evolution of Δ RGW is calculated by coupling the static TF-EP framework (this study) with the hydrological model driven by the combined RCP-SSP forcings. As expected, there is a declining trend of Δ RGW due to increased water demand and decreased surface water availability projected into the future. Different GCMs behave very differently and the variability across them increases dramatically at the end of 2030s. This highlights the need to use an ensemble-based framework to address the uncertainties. Although we did not include Figure R5 in the revised manuscript, we highlight the importance of considering these issues in Lines 330-337 on Page 18: “Although our TF-EP framework does not explicitly include the temporal dimension, we show how trade-off pathways vary with water availability, which indeed vary over time. Further work is needed to extend this static framework to a dynamic version in the temporal space and consider future uncertainties of both water supply and demand due to changing climate and human interventions. Such challenges can be tackled through robust decision making (RDM) approaches combined with model-based large ensembles driven by different climate and social-economic scenarios (He et al., 2019).”

Figure R5. Percentage change of recovered groundwater (ΔRGW) during 2016-2030. RGW is defined as the optimal GW abstraction difference between current and future SWE (see the inset schematic diagram). For year t , percentage change of RGW ($\Delta\text{RGW}(t)$) is calculated as the percentage difference of RGW between the baseline (historical) and SSP (Shared Socioeconomic Pathways) scenario. Climate projections are taken from four Global Climate Models under RCP6.0 scenario. Projections of future water demand from domestic, industrial, and livestock sectors are obtained from the Water Futures and Solutions (WFaS) Initiative (Wada et al., 2016) under the SSP2 (middle-of-the-road scenario indicating a medium-level of adaptation and mitigation) scenario.

Sensitivity results of EP to increased irrigation efficiency are shown in Figure R6 (also included in the Supplementary Material as Figure S1). As expected, increased irrigation efficiency (dashed lines) can reduce groundwater abstraction (EP is shifted to right). But this is only true when water is limited (lower left corner) and only under lower penetration of SWE (yellow color). Our results also highlight the dominant role of SWE, as under 40% penetration of SWE, increased irrigation efficiency does not affect the EP (red color). Discussion on this can be found in Lines 323-330 on Pages 17-18: “Nevertheless, the proposed framework is flexible enough to integrate other factors into the optimization framework, which can be done by including additional constraints. Considering improved technologies as an example. While future irrigation efficiency is expected to increase by 17-22% in California (Cooley et al., 2014), this only has limited effects on the trade-off pathways under current penetration of SWE when water is limited (see Figure S1). With higher penetration of SWE, this limited effect disappears regardless of water availability conditions, highlighting the dominant role of SWE compared to improved irrigation efficiency.”

Figure R6. Sensitivity tests showing optimal expansion paths under current (yellow lines, 17%) and future (red lines, 40%) penetration of SWE considering changes in irrigation efficiency. Dashed lines show results considering decreased irrigation water requirement (IWR), given that future irrigation efficiency in California is expected to increase by 17-22% in all year types (i.e., normal, dry, wet) (Cooley et al., 2014). Note: Under 40% penetration of SWE, red dashed lines and red solid lines overlap with each other, which indicate that increased irrigation efficiency does not influence expansion paths when penetration of SWE is high.

Trade-offs between economic, engineering and social benefits are relevant when managing nexus systems, yet this paper reports on efficient combinations of hydropower generation and groundwater use. The paper describes a framework for nexus systems and sustainable development goals (SDGs) but it isn't clear how groundwater use constitutes an SDG and whether trade-offs between economic production (hydropower) and individual supply sources (groundwater in this paper) are relevant to policy makers.

Response: The reviewer is right that SDGs do not have groundwater indicator explicitly. Only water scarcity is an indicator in SDGs. Also pointed out by Reviewer 1, our framework may not be directly applied to analyze the trade-offs between different SDGs. Our intention was to note that the framework could be useful in other contexts including analysis of trade-offs between SDGs, which is vital to understand because of how intertwined the goals are. Nevertheless, we agree that our statements may be premature, and so we have deleted them in the abstracts as well as the discussion section to tone down the conclusion.

3. Responses to Reviewer #3

The submitted manuscript proposes a trade-off frontier framework to account for the role of solar and wind energy in the water-food-energy nexus. The idea is interesting, innovative and promising, but the manuscript fails in presenting it in a clear and intelligible way. A lot of confusion arises when describing the trade-off frontier (TF) – actually there is no explanation supporting the construction of the TF curve. Also, both figures are extremely dense, the caption is poor as well as their description in the main text. I strongly encourage the authors to significantly re-define the manuscript structure and better clarify their main messages and outcomes. As it stands, I cannot recommend this paper to be published in Nature Communications its present form.

Response: We appreciate the reviewers' comments to improve the clarity of our manuscript. We have provided more details in the Method Section on how to construct the TF curve (Page 24). In addition, figure captions have been revised to include more detailed description. In our initial submission, the main messages and outcomes of our paper are already summarized as a title for each section. In the revision, we have integrated and re-emphasized these findings in the last paragraph (Lines 376-383 on Page 20): “In this study, we find that combining SWE within hydropower systems may achieve an under-appreciated mutual benefit for the water-food-energy nexus. Here, by quantifying the water sustainability value of SWE using a trade-off frontier framework, our case study in California has revealed the ability of increased SWE penetration to enhance drought resilience and groundwater sustainability. The resulting co-benefit on groundwater sustainability can further relieve the impact of groundwater regulations on agricultural revenue loss.”

Some comments:

p. 6 l. 98: authors should explain how water surface constraints vary from dry to wet in Fig 1

Response: The water constraints in Figure 1 indicate surface water availability, which are estimated from the physical hydrological model averaged over dry, normal, and wet years during the historical period. With increased surface water availability (from a dry to wet year), more water can be allocated for hydropower generation and irrigation, therefore the constraint moves towards the upper-right direction. We have added further explanation on this in the caption of Figure 1 (Page 22).

p. 6 l. 104: where “inefficient/unattainable” status could be identified in TF? Please add an explanation

Response: In Figure 1, the TF curves (grey solid lines) represent efficient water allocation strategies. Inefficient status is where the water allocation moves downwards of the TF curve and indicates that surface water is not fully used for hydropower generation and irrigation. In contrast, if water allocation lies above the TF curve, then this strategy is unattainable, because there is not enough water to satisfy the needs of hydropower and irrigation. This is explained on Pages 7-8 from Line 106 to 111: “A strategy is inefficient if surface water is not fully used

for hydropower (production is lower than potential) and agriculture (irrigation is less than crop demand), and groundwater is used for irrigation instead. A strategy is unattainable if the water demand for both hydropower production and irrigation exceeds the surface water availability, and the shortfall in irrigation demand cannot be satisfied by the current groundwater abstraction rate.”

However, to make the explanation clear, we revised the sentence in Lines 103-106 on Page 7: “Given the water constraint in a certain year, surface water allocation strategies are efficient if they fall along the TF curve, while they are inefficient/unattainable if strategies fall below/above the TF.”

p.7 ll. 116-118: The authors state that the black point in Fig 1 represents the condition such that maximum revenue and efficient water allocation are met. Why maximum revenue? The iso-revenue line shows equal revenues for different groundwater and hydropower conditions... I found this part quite obscure

Response: The reviewer is correct that the iso-revenue line shows equal revenues for different combinations of groundwater and hydropower conditions. However, this is only true if there are no constraints. If we add water availability constraints, then only the tangent point between the constraint and the iso-revenue curve will give the maximum revenue. This can be demonstrated using the following schematic diagram (Figure R7).

Figure R7. Maximizing total revenue given surface water constraint.

In our study, the objective is to maximize the total revenue (R) by allocating surface water between hydropower and irrigation. At point A, the constraint line (surface water availability line, blue color) and the iso-revenue curve R_2 (red line) are tangent, and therefore no higher level of revenue (e.g., strategy D) can be attained. If we choose other strategies, such as point B and C, then we will obtain lower revenue as the points lie along the iso-revenue line R_1 , which is smaller than R_2 . This is only an intuitive demonstration. Strict mathematical proof can be found in Pindyck and Rubinfeld (2013). Microeconomics, 8th edition.

p. 7 l. 123: inwards and outwards: what is the reference?

Response: The reference is the normal year. We have made this clear in Lines 123-126 on Page 8: “On top of these factors, hydroclimate variability will shift the TF inwards and outwards for low (lower surface water availability in a dry year) and high inflow (higher surface water availability in a wet year) conditions, respectively, compared to the normal year”.

p. 8 ll. 142-143: why do the current scenario shows a smaller curvature of iso-revenue lines compared to future scenario? Also, what is the % of hydropower production now? And in 2030?

Response: The reason why the current scenario shows a smaller curvature compared to future scenario is explained in Lines 146-148 on Page 9: “This implies that as more SWE is deployed and the hydroelectricity price goes down, to maintain the same revenue, one unit of abstraction of groundwater requires more hydroelectric generation to compensate the pumping cost.”

We do not know the percentage of hydropower production in 2030 yet, but we assume it will be similar to or slightly higher than the current percentage given that large sites for hydropower plants have already been taken, and presumably there will be more efficient turbines. Moreover, the percentage production of hydropower highly depends on the water availability. As we do not know the hydroclimate of 2030, it is hard to provide an estimate.

p. 20 eq 1: is there an error in the 3rd water constraint (maybe a sum instead of a product)?

Response: No, there is no error. The 3rd water constraint does not have a product sign. Instead, it is a comma, meaning that both s_w^{Hydro} and s_w^{Crop} should be within 0 and s_w . To avoid any misunderstanding, we have rewritten this constraint on two separate lines on Page 24.

Figs S1-S8: a detailed explanation of plots and tables should be added

Response: We have included more detailed explanation into the captions of Figs S4-S19 in the revised supplementary material.

REVIEWERS' COMMENTS:

Reviewer #1 (Remarks to the Author):

The authors have adequately addressed reviewer comments and made clarifications to improve the manuscript. I have no additional review comments.

Reviewer #2 comments:

This paper covers the interesting topic of the food-energy-water nexus in California and its optimization to inform groundwater policy, and in particular to evaluate the increased use of renewable energy sources (pv, wind). The reviewer has identified the following issues:

It wasn't clear what the spatial structure of the analysis is. A calibrated hydrological model at several stations is provided, but it isn't clear how this is connected to California's distributed water and energy system. For example what is the spatial unit of analysis for groundwater? The paper describes using groundwater parameter data from Knapp et al. 2003, but that study was just for Kern county, i.e., a relatively small region within California. California's Central Valley has several distinct aquifers some unconfined other confined; are these represented in this groundwater themed analysis?

The authors noted that they did not account for spatial variability in the distributed energy system directly. They added a discussion of the limitations in their approach and how future work might address these. They also recalculated groundwater to avoid over-dependence on Kern County results and included extensive sensitivity analyses in supplementary material to illustrate how assumptions may affect the results.

Are urban water demands represented? And if so at what spatial scale, and how are these demands connected to California's water supply network?

The authors answer the question in their response, but did not change the manuscript in response. Thus, the question could still arise among readers.

California's water management uses infrastructure – both for transmission (transfers) and storage (reservoirs). How water is managed and allocated via this infrastructure determines the benefits obtained from water by the different sectors. Also, how water is stored and allocated in different times of the year, via water rights and water markets, determines the economic benefits from water use. Particularly under scarcity water markets become active and determine the economic costs of scarcity. Finally, in California groundwater and surface water are used conjunctively in "water banks". Is the reviewer correct that these essential aspects of California's water management have been ignored in this study? If so, is there a danger of making policy recommendations about a hydrologic-engineered-economic system without considering with sufficient accuracy how water supply and demand manifest over space and time.

The authors addressed this comment by acknowledging limitations, as they did for the first comment of this reviewer.

The optimization is monetary; does that imply environmental, engineering and social aspects are not considered? For example, how is supply reliability monetized in this framework?

The authors again acknowledge the limits of their study and add comments about how the value of additional storage might be a topic for future research.

The authors have applied a form of annual dynamic optimization but results are not given over time, so it is not possible for readers to evaluate how the model represents the system's evolution over time. Does water use change over time? Do water users learn or evolve? Do farmers make decisions about crop choice or irrigation technologies? The paper speaks of finding an optimal pathway (over time the reviewer assumes) yet recent literature on development pathways underscores the importance of considering the uncertainty of both future supply and demand which doesn't seem to be considered in this analysis. Does the annual dynamic economic optimization only consider one scenario, the historical one?

Explanation that the approach was not explicit in time but rather used years with different water availability was included in the revised manuscript. The authors also noted how a dynamic analysis can be done and pointed to a paper that the team gave at an AGU meeting.

Trade-offs between economic, engineering and social benefits are relevant when managing nexus systems, yet this paper reports on efficient combinations of hydropower generation and groundwater use. The paper describes a framework for nexus systems and sustainable development goals (SDGs) but it isn't clear how groundwater use constitutes an SDG and whether trade-offs between economic production (hydropower) and individual supply sources (groundwater in this paper) are relevant to policy makers.

The material on SDGs was deleted.

Reviewer #3 (Remarks to the Author):

Thanks for addressing my comments. I am pleased with this reply and have no further comments to make.

2. Responses to Reviewer #2

Are urban water demands represented? And if so at what spatial scale, and how are these demands connected to California's water supply network?

The authors answer the question in their response, but did not change the manuscript in response. Thus, the question could still arise among readers.

Response: We have included our previous response into the Supplementary Note 2: "Urban water demand is linked to population densities, which are connected to the urban water supply network. In our coupled hydro-economic optimization framework, we assume that pixel-level urban water demands are satisfied first. The remaining available surface water at the pixel level is then aggregated to the whole California. This aggregated total surface water availability is used as the constraint in the optimization model."